# COVID-19 In-Hospital Mortality in People with Diabetes Is Driven by Comorbidities and Age—Propensity Score-Matched Analysis of Austrian National Public Health Institute Data

**DOI:** 10.3390/v13122401

**Published:** 2021-11-30

**Authors:** Faisal Aziz, Felix Aberer, Alexander Bräuer, Christian Ciardi, Martin Clodi, Peter Fasching, Mario Karolyi, Alexandra Kautzky-Willer, Carmen Klammer, Oliver Malle, Erich Pawelka, Thomas Pieber, Slobodan Peric, Claudia Ress, Michael Schranz, Caren Sourij, Lars Stechemesser, Harald Stingl, Hannah Stöcher, Thomas Stulnig, Norbert Tripolt, Michael Wagner, Peter Wolf, Andreas Zitterl, Alexander Christian Reisinger, Jolanta Siller-Matula, Michael Hummer, Othmar Moser, Dirk von-Lewinski, Philipp Eller, Susanne Kaser, Harald Sourij

**Affiliations:** 1Interdisciplinary Metabolic Medicine Trials Unit, Department of Endocrinology and Diabetology, Medical University of Graz, 8036 Graz, Austria; faisal.aziz@stud.medunigraz.at (F.A.); felix.aberer@medunigraz.at (F.A.); oliver.malle@medunigraz.at (O.M.); hannah.stoecher@stud.medunigraz.at (H.S.); norbert.tripolt@medunigraz.at (N.T.); 2Medical Division of Endocrinology, Rheumatology and Acute Geriatrics, Hospital Ottakring, 1160 Vienna, Austria; alexander.braeuer@wienkav.at (A.B.); peter.fasching@gesundheitsverbund.at (P.F.); 3Clinical Division for Internal Medicine, Endocrinology, Diabetology and Metabolic Diseases, St. Vinzenz Hospital Zams, 6511 Zams, Austria; christian.ciardi@krankenhaus-zams.at; 4Clinical Division for Internal Medicine, Konvent Hospital Barmherzige Brüder Linz, 4020 Linz, Austria; martin.clodi@bblinz.at (M.C.); carmen.klammer@bblinz.at (C.K.); 54th Medical Division with Infectiology, SMZ Süd—KFJ-Hospital Vienna, 1100 Vienna, Austria; mario.karolyi@wienkav.at (M.K.); erich.pawelka@wienkav.at (E.P.); 6Division for Endocrinology and Metabolism, Medical University of Vienna, AKH, 1090 Vienna, Austria; alexandra.kautzky-willer@meduniwien.ac.at (A.K.-W.); peter.wolf@meduniwien.ac.at (P.W.); 7Division of Endocrinology and Diabetology, Medical University of Graz, 8036 Graz, Austria; thomas.pieber@medunigraz.at; 8Department of Medicine III and Karl Landsteiner, Institute for Metabolic Diseases and Nephrology, Clinic Hietzing, Vienna Health Care Group, 1130 Vienna, Austria; slobodan.peric@gesundheitsverbund.at (S.P.); thomas.stulnig@gesundheitsverbund.at (T.S.); andreas.zitterl@wienkav.at (A.Z.); 9Department for Internal Medicine I, Medical University of Innsbruck, 6020 Innsbruck, Austria; claudia.ress@i-med.ac.at; 10Department for Inner Medicine, Paracelsus-Private Medical University, 5020 Salzburg, Austria; m.schranz@salk.at (M.S.); l.stechemesser@salk.at (L.S.); 11Division of Cardiology, Medical University of Graz, 8036 Graz, Austria; caren.sourij@medunigraz.at (C.S.); dirk.von-lewinski@medunigraz.at (D.v.-L.); 12Clinical Division for Internal Medicine, Hospital Melk, 3390 Melk, Austria; harald@stingl.info (H.S.); michael.c.wagner@gmx.at (M.W.); 13Intensive Care Unit, Department of Internal Medicine, Medical University of Graz, 8036 Graz, Austria; alexander.reisinger@medunigraz.at (A.C.R.); philipp.eller@medunigraz.at (P.E.); 14Division of Cardiology, Medical University of Vienna, AKH, 1090 Vienna, Austria; jolanta.siller-matula@meduniwien.ac.at; 15Austrian National Public Health Institute, 1010 Vienna, Austria; michael.hummer@goeg.at; 16Department of Exercise Physiology & Metabolism, Institute of Sports Science, University of Bayreuth, 95445 Bayreuth, Germany; Othmar.Moser@uni-bayreuth.de

**Keywords:** COVID-19, diabetes, intensive care, mortality, SARS-CoV-2

## Abstract

Background: It is a matter of debate whether diabetes alone or its associated comorbidities are responsible for severe COVID-19 outcomes. This study assessed the impact of diabetes on intensive care unit (ICU) admission and in-hospital mortality in hospitalized COVID-19 patients. Methods: A retrospective analysis was performed on a countrywide cohort of 40,632 COVID-19 patients hospitalized between March 2020 and March 2021. Data were provided by the Austrian data platform. The association of diabetes with outcomes was assessed using unmatched and propensity-score matched (PSM) logistic regression. Results: 12.2% of patients had diabetes, 14.5% were admitted to the ICU, and 16.2% died in the hospital. Unmatched logistic regression analysis showed a significant association of diabetes (odds ratio [OR]: 1.24, 95% confidence interval [CI]: 1.15–1.34, *p* < 0.001) with in-hospital mortality, whereas PSM analysis showed no significant association of diabetes with in-hospital mortality (OR: 1.08, 95%CI: 0.97–1.19, *p* = 0.146). Diabetes was associated with higher odds of ICU admissions in both unmatched (OR: 1.36, 95%CI: 1.25–1.47, *p* < 0.001) and PSM analysis (OR: 1.15, 95%CI: 1.04–1.28, *p* = 0.009). Conclusions: People with diabetes were more likely to be admitted to ICU compared to those without diabetes. However, advanced age and comorbidities rather than diabetes itself were associated with increased in-hospital mortality in COVID-19 patients.

## 1. Introduction

Severe acute respiratory syndrome coronavirus 2 (SARS-CoV-2) is a novel virus that caused a pandemic of coronavirus disease starting in 2019 (COVID-19). As of 27 July 2021, 194.72 million confirmed cases of COVID-19 and 4.17 million deaths have been reported globally with an estimated fatality rate of approximately 2% [1,2].

Earlier research from China, the United States, and Europe has reported a high prevalence of diabetes (17–33%) among people hospitalized for COVID-19 [3,4,5]. In addition, people with diabetes are more likely to develop severe SARS-CoV-2 infection and complications such as diabetic ketoacidosis and death [2,6,7]. Various mechanisms have been delineated to understand the relationship between diabetes and COVID-19. As such, people with diabetes have a compromised immune system and dysfunctional inflammatory response, which may result in a more complex course of the disease and prolonged recovery. Moreover, SARS-CoV-2 infection may disrupt the regulation of blood glucose in people with diabetes, which provides a thriving environment to the virus, thereby making it challenging to treat this infection [8].

Recent studies have shown that certain risk factors such as old age, sex, and smoking status and the presence of comorbidities such as cardiovascular disease, dementia, liver disease, renal disease, and cancer also significantly increase the risk of developing severe disease and mortality in people with COVID-19 [5,7,9,10]. In particular, type 2 diabetes is a disease of advanced age with a high burden of multimorbidity [11]; it is speculated that the co-existence of these underlying conditions exacerbates the prognosis of COVID-19 in people with diabetes [12]. However, evidence concerning the impact of diabetes on COVID-19 outcomes that is independent of age, sex, and multimorbidity is still emerging [13,14]. Therefore, this study assessed the association of diabetes with the severity of disease leading to intensive care unit (ICU) admission and in-hospital mortality in patients hospitalized for SARS-CoV-2 infection in Austria by adopting a propensity score matching (PSM) method to account for sex, age, and comorbidities.

## 2. Materials and Methods

### 2.1. Study Design and Data Source

The ‘Strengthening the Reporting of Observational Studies in Epidemiology (STROBE)’ checklist was used for reporting this study [15]. A retrospective cohort study was conducted in patients hospitalized for primary and secondary SARS-CoV-2 infection in Austria. These data were collected by the “Data platform COVID-19” provided by the Austrian National Public Health Institute (Gesundheit Österreich GmbH, Stubenring 6, 1010, Vienna, Austria).

This platform captures countrywide epidemiological and clinical data of COVID-19 patients in Austria to improve understanding and provide updated evidence regarding SARS-CoV-2 infection in the country. The details about the Austrian data platform are available at: https://datenplattform-covid.goeg.at/english, accessed on 3 May 2021.

### 2.2. Data Extraction and Study Variables

This study includes data of COVID-19 patients with and without diabetes mellitus who were hospitalized between March 2020 and March 2021 in Austria. Of the 51,469 patients recorded in the database, 40,602 patients were included in the final analysis after excluding patients aged below 20 years and with missing and duplicate data (Figure 1). Study variables comprised age deciles at admission, sex, number of hospitalizations for COVID-19, geographic regions, diabetes mellitus, comorbidities, ICU admission, and in-hospital mortality.

Diabetes mellitus was defined as per the International Classification of Disease (ICD) version 10 codes of E10, E11, E12, E13, and E14. Comorbidities were identified by referring to Charlson and Elixhauser indices and defined as per the ICD-10 codes. The comorbidities included myocardial infarction, cardiac arrhythmias, valvular heart disease, hypertension, congestive heart failure (CHF), peripheral vascular disease, stroke, chronic obstructive pulmonary disease (COPD), pulmonary circulation disorders, dementia, rheumatoid disease, peptic ulcer disease, liver disease, paralysis, other neurological disorders, chronic renal disease, cancer with/without metastasis, Human Immunodeficiency Virus (HIV)/Acquired Immune Deficiency Syndrome (AIDS), hypothyroidism, coagulopathy, fluid and electrolyte disorders, blood loss anemia, deficiency anemia, alcohol abuse, psychosis, and depression [16,17]. The detailed description of diseases included as comorbidities with their respective ICD-10 codes is provided in Appendix A. In addition to individual comorbidities, the Charlson Comorbidity Index (CCI) was calculated using the ‘comorbidity’ package to measure the cumulative impact of comorbidities (except diabetes) on outcomes.

### 2.3. Ethical Considerations

The study was reviewed by the ethics committee of the Medical University of Graz, Graz, Austria (EK 32-355 ex 19/20) and conformed to the 1964 declaration of Helsinki and guidelines of the International Conference on Harmonization for Good Clinical Practice (ICH GCP E6 guidelines). As it was a retrospective analysis of anonymized data, no informed consent was obtained from the patients.

### 2.4. Statistical Analysis

Data were received in the Microsoft Excel format and analyzed in R version 4.1.0. Qualitative variables were summarized as frequencies with corresponding percentages (%) and compared with diabetes status using Pearson’s chi-square or Fischer’s exact tests and standardized mean differences (SMD).

The impact of diabetes on both ICU admission and in-hospital mortality was assessed using unmatched and PSM logistic regression. In unmatched logistic regression, the unadjusted association of diabetes with both outcomes was ascertained in the entire cohort (N = 40,602). In PSM analysis, the logistic regression model of diabetes with unbalanced variables having an SMD of ≥0.1 each (Table 1) was fitted to estimate the propensity score. Next, the ‘MatchIt’ package was used to generate a 1:1 without replacement PSM cohort of diabetes (n = 4971) and non-diabetes (n = 4971) patients by applying the nearest neighbor method. After propensity score matching, the SMD was re-estimated in the PSM cohort to determine whether the balance was achieved across selected variables or not. The results of both unmatched and PSM logistic regression were reported as odds ratios (OR) with corresponding 95% confidence intervals (CI) and *p*-values. Furthermore, the association of diabetes with outcomes was adjusted for variables that remained unbalanced even after performing the PSM. A *p*-value of <0.05 was chosen to determine statistical significance.

## 3. Results

Table 1 shows the distribution of characteristics and comorbidities of hospitalized COVID-19 patients in unmatched and PSM cohorts. In the unmatched cohort, two-thirds of patients were 60 to 89 years old and 52.4% were men. Common comorbidities were cardiovascular disease (33.1%), hypertension (26%), chronic renal disease (7.8%), COPD (5.6%), and dementia (5.0%). Of the total patients, 18.4% had a CCI score of 1–2, 5.4% had 3–4, and 1.4% had a score of ≥5.

A total of 4971 (12.2%) hospitalized patients had diabetes, 5968 (14.7%) were admitted to ICU, and 6569 (16.2%) died in the hospital. Patients with diabetes were older (*p* < 0.001), included more males (57.5% vs. 51.7%, *p* < 0.001), and had a higher ICU admission rate (18.3% vs. 14.2%, *p* < 0.001) and in-hospital mortality (18.8% vs. 15.8%, *p* < 0.001) than those without diabetes. Among comorbidities, CVD (78.1% vs. 26.9%, *p* < 0.001), COPD (11.5% vs. 4.8%, *p* < 0.001), dementia (9.7% vs. 4.4%, *p* < 0.001), liver disease (7.2% vs. 1.9%, *p* < 0.001), chronic renal disease (21.7% vs. 5.9%, *p* < 0.001), cancer (4.8% vs. 3.0%, *p* < 0.001), hypothyroidism (6.9% vs. 2.8%, *p* < 0.001), fluid and electrolyte disorders (6.1% vs. 3.2%, *p* < 0.001), and depression (5.0% vs. 2.3%, *p* < 0.001) were significantly more prevalent in patients with diabetes than those without diabetes. Patients with diabetes had significantly higher (*p* < 0.001) CCI scores compared to patients without diabetes. In the PSM cohort of 4971 patients with diabetes and 4971 matched patients without diabetes, characteristics and comorbidities were well balanced for 13 variables as indicated by the SMD of <0.10 (Table 1).

Table 2 shows the comparison and logistic regression analysis of ICU admission with diabetes and other variables for both unmatched and PSM cohorts. In the unmatched analysis, patients with diabetes were 36% more likely (OR: 1.36, 95%CI: 1.25–1.47, *p* < 0.001) to be admitted to ICU compared to patients without diabetes. This association was attenuated by more than half (OR: 1.15, 95%CI: 1.04–1.28, *p* = 0.009) in the PSM analysis, nonetheless remaining significant. In addition, males and older patients were more likely to be admitted to ICU in both unmatched and PSM analyses. The odds of ICU admission were significantly higher in patients having a myocardial infarction, CHF, respiratory disorders, and coagulation disorders compared to those without these comorbidities in both unmatched and PSM analyses. A CCI score of 3–4 showed a significant association with ICU admission in the PSM analysis.

Table 3 shows the comparison and logistic regression analysis of in-hospital mortality with diabetes and other variables for both unmatched and PSM cohorts. The unmatched analysis showed 24% higher odds of mortality (OR: 1.24, 95%CI: 1.15–1.34, *p* < 0.001) in people with diabetes compared to those without diabetes. The association of diabetes with in-hospital mortality (OR: 1.08, 95%CI: 0.97–1.19, *p* = 0.146) became insignificant in the PSM analysis. Although hypertension was not well balanced between diabetes and non-diabetes patients, adjusting for it did not change the magnitude of association between diabetes and in-hospital mortality significantly (OR: 1.09, 95%CI: 0.99–1.21, *p* = 0.065). In both unmatched and PSM analysis, the odds of mortality increased with age and CCI score and were significantly higher in males and those admitted to ICU. Patients with myocardial infarction, cardiac arrhythmias, valvular heart disease, congestive heart failure, peripheral artery disease, or stroke had higher odds of mortality (*p* < 0.001) compared to those without these CVDs. Likewise, patients with COPD, pulmonary circulation disorders, dementia, chronic renal disease, cancer, and hypothyroidism had higher odds of mortality than those without these comorbidities.

## 4. Discussion

This countrywide retrospective study examined the impact of diabetes on the severity of the infection and in-hospital mortality in patients hospitalized for COVID-19. The analysis showed that hospitalized COVID-19 patients with diabetes were older and had a higher burden of multimorbidity and CCI score compared to those without diabetes. Patients with diabetes were more likely to have severe COVID-19 disease than those without diabetes as evidenced by higher odds of ICU admission even after propensity matching of characteristics, comorbidities, and the CCI score. However, diabetes was not independently associated with in-hospital mortality after matching diabetes and non-diabetes cohorts. While old age, male sex, and comorbidities were significantly associated with both outcomes in unmatched and matched analyses.

This study adopted the PSM method to assess the independent impact of diabetes on the severity of COVID-19 and in-hospital mortality. The association of diabetes with mortality was attenuated threefold i.e., from 24% in the unmatched cohort to 8% in the PSM cohort. These findings suggest that diabetes has no independent impact on in-hospital mortality in COVID-19 patients. Instead, advanced age, sex, and multimorbidity are responsible for the previously shown association between diabetes and in-hospital mortality. Our findings are consistent with a previous study that employed the PSM analysis and found no significant association between diabetes and in-hospital mortality in French patients hospitalized for COVID-19 [14]. In contrast, most studies have reported a significant positive association between diabetes and COVID-19 related mortality. For instance, a recent meta-analysis has demonstrated a significant association between diabetes and COVID-19 mortality that attenuated after adjusting for age and comorbidities [18]. Another meta-analysis of 31,067 patients has also reported a higher COVID-19 related mortality in patients with diabetes [19]. Similarly, a large study in England reported 80% higher odds of COVID-19 deaths in people with type 2 diabetes than those without diabetes [20]. Likewise, two small studies from China found diabetes to be a significant risk factor of mortality in COVID-19 patients even after adjusting for comorbidities and laboratory markers [21,22]. Even though previous studies have demonstrated a significant association between diabetes and COVID-19 mortality, most studies and those included in meta-analyses have not adjusted for various significant comorbidities and risk factors. These limitations further support our finding regarding the underlying impact of age, sex, and multimorbidity on COVID-19 mortality in patients with diabetes.

In this study, patients with diabetes were 36% more likely to have severe COVID-19 disease as represented by ICU admission in the unmatched analysis that decreased to 15% in the PSM analysis but remained significant. These findings are consistent with previous studies; however, the direct comparison across studies is not possible because of differences in defining criteria of the severity of COVID-19 disease and statistical analysis methods across studies. Moreover, the degree of association between diabetes and the severity of COVID-19 disease in our study is much lower than in other studies. For instance, meta-analyses performed in different phases of the COVID-19 pandemic have reported the pooled odds of developing severe COVID-19 disease ranging from 1.58 to 2.75 in people with diabetes compared to non-diabetes [23,24,25,26]. The vast heterogeneity in the degree of association between diabetes and COVID-19 severity is primarily due to different adjustment factors like age, obesity, smoking, and comorbidities considered by studies [24,27]. Moreover, previous studies have shown that glucose-lowering medication might be associated with COVID-19 outcomes [28]. However, we could not adjust for medication in our study due to the unavailability of treatment regimen information. Of note, dementia was associated with in-hospital mortality, while it was significantly associated with lower ICU admission. We believe that this association is rather due to a selection bias of patients being admitted to ICU than a biological association.

Our findings must be interpreted with some limitations. As common in healthcare data, there is a possibility of miscoding or under-coding of diagnoses in this database. Therefore, diabetes was not classified into subtypes because of the uncertainty in the ICD coding accuracy of this diagnosis. Moreover, laboratory measurements of glucose and other biochemical parameters were not available for all patients and therefore not included in the analysis. This limitation might have resulted in residual confounding between diabetes and COVID-19 outcomes. Furthermore, in-hospital mortality was defined as death from any underlying cause, which might have captured deaths resulting from causes other than COVID-19. However, considering that this database holds data only on patients with COVID-19, the likelihood of including non-COVID-19-related deaths is low. In addition, we only analyzed in-hospital mortality in this study, however previous studies have shown that the pandemic might impact overall mortality in people with diabetes as well [29]. Last, our cohort comprised Austrian citizens only, hence the findings might not be generalized to other populations with different ethnicities.

## 5. Conclusions

This countrywide retrospective cohort analysis of patients hospitalized for COVID-19 found that people with diabetes had an advanced age and a higher burden of multimorbidity compared to those without diabetes. Consistent with the existing research, hospitalized patients with diabetes were significantly more likely to suffer from severe COVID-19 illness compared to those without diabetes; whereas, comorbidities and old age rather than diabetes per se were responsible for the increased likelihood of COVID-19 related in-hospital mortality. This study has confirmed that advanced age and comorbidities play a major role in both COVID-19 disease severity and death even in patients with diabetes and hence should be considered in risk stratification, management of patients, and national vaccination strategies.

## Figures and Tables

**Figure 1 viruses-13-02401-f001:**
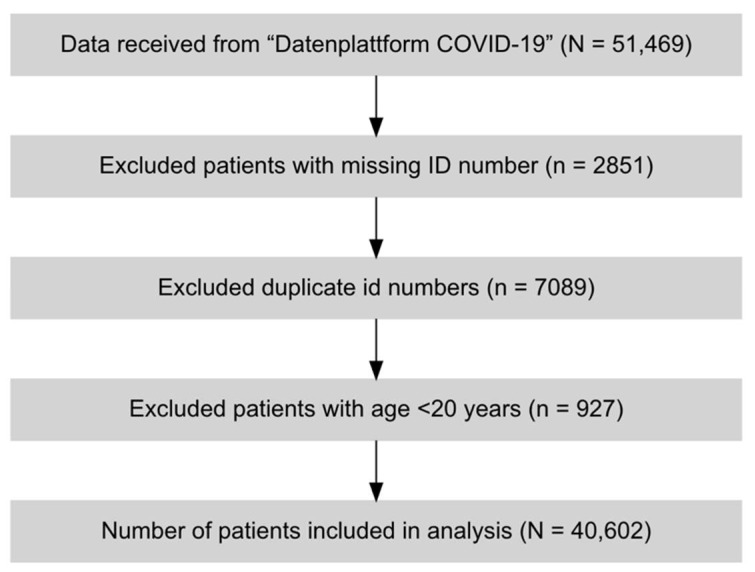
Flow diagram of extraction.

**Table 1 viruses-13-02401-t001:** Characteristics and comorbidities of diabetes and non-diabetes patients hospitalized for COVID-19 patients in unmatched and propensity score matched cohorts.

Variables	Unmatched Cohort					PSM Cohort		
	All	No Diabetes n (%)	Diabetes n (%)	SMD	*p*-Value	No Diabetes n (%)	Diabetes n (%)	SMD
N	40,602	35,631 (87.8)	4971 (12.2)			4971 (50.0)	4971 (50.0)	
Men	21,278 (52.4)	18,419 (51.7)	2859 (57.5)	0.117	<0.001	2839 (57.1)	2859 (57.5)	0.008
Women	19,324 (47.6)	17,212 (48.3)	2112 (42.5)			2132 (42.9)	2112 (42.5)	
Age categories								
20–29	1057 (2.6)	1050 (2.9)	7 (0.1)	0.403	<0.001	5 (0.1)	7 (0.1)	0.028
30–39	1576 (3.9)	1536 (4.3)	40 (0.8)			44 (0.9)	40 (0.8)	
40–49	2479 (6.1)	2306 (6.5)	173 (3.5)			177 (3.6)	173 (3.5)	
50–59	5363 (13.2)	4751 (13.3)	612 (12.3)			643 (12.9)	612 (12.3)	
60–69	6674 (16.4)	5743 (16.1)	931 (18.7)			931 (18.7)	931 (18.7)	
70–79	9664 (23.8)	8133 (22.8)	1531 (30.8)			1490 (30.0)	1531 (30.8)	
80–89	10,532 (25.9)	9144 (25.7)	1388 (27.9)			1387 (27.9)	1388 (27.9)	
90+	3257 (8.0)	2968 (8.3)	289 (5.8)			294 (5.9)	289 (5.8)	
ICU admission	5968 (14.7)	5057 (14.2)	911 (18.3)	0.110	<0.001	812 (16.3)	911 (18.3)	0.053
In-hospital mortality	6569 (16.2)	5632 (15.8)	937 (18.8)	0.080	<0.001	881 (17.7)	937 (18.8)	0.029
Comorbidities								
Myocardial infarction	340 (0.8)	236 (0.7)	104 (2.1)	0.123	<0.001	107 (2.2)	104 (2.1)	0.004
Cardiac arrhythmias	4235 (10.4)	3134 (8.8)	1101 (22.1)	0.376	<0.001	1217 (24.5)	1101 (22.1)	0.055
Valvular heart disease	990 (2.4)	738 (2.1)	252 (5.1)	0.162	<0.001	307 (6.2)	252 (5.1)	0.048
Hypertension	10,538 (26.0)	7209 (20.2)	3329 (67.0)	1.069	<0.001	3002 (60.4)	3329 (67.0)	0.137
CHF	2011 (5.0)	1381 (3.9)	630 (12.7)	0.323	<0.001	602 (12.1)	630 (12.7)	0.017
PVD	1341 (3.3)	855 (2.4)	486 (9.8)	0.312	<0.001	385 (7.7)	486 (9.8)	0.072
Stroke	1212 (3.0)	834 (2.3)	378 (7.6)	0.244	<0.001	378 (7.6)	378 (7.6)	0.002
COPD	2271 (5.6)	1699 (4.8)	572 (11.5)	0.248	<0.001	568 (11.4)	572 (11.5)	0.003
Pulmonary circulation disorders	591 (1.5)	474 (1.3)	117 (2.4)	0.076	<0.001	129 (2.6)	117 (2.4)	0.016
Dementia	2050 (5.0)	1570 (4.4)	480 (9.7)	0.206	<0.001	488 (9.8)	480 (9.7)	0.005
Rheumatoid disease	254 (0.6)	198 (0.6)	56 (1.1)	0.063	<0.001	77 (1.5)	56 (1.1)	0.037
Peptic ulcer disease	86 (0.2)	62 (0.2)	24 (0.5)	0.054	<0.001	20 (0.4)	24 (0.5)	0.012
Liver disease	1031 (2.5)	672 (1.9)	359 (7.2)	0.258	<0.001	305 (6.1)	359 (7.2)	0.044
Paralysis	126 (0.3)	98 (0.3)	28 (0.6)	0.045	0.001	34 (0.7)	28 (0.6)	0.015
Other neurological disorders	1071 (2.6)	839 (2.4)	232 (4.7)	0.126	<0.001	201 (4.0)	232 (4.7)	0.031
Renal disease	3173 (7.8)	2092 (5.9)	1081 (21.7)	0.473	<0.001	1051 (21.1)	1081 (21.7)	0.015
Cancer	1318 (3.2)	1079 (3.0)	239 (4.8)	0.092	<0.001	229 (4.6)	239 (4.8)	0.009
Hypothyroidism	1335 (3.3)	993 (2.8)	342 (6.9)	0.192	<0.001	347 (7.0)	342 (6.9)	0.004
Coagulation disorders	252 (0.6)	179 (0.5)	73 (1.5)	0.098	<0.001	46 (0.9)	73 (1.5)	0.050
Fluid and electrolyte disorders	1433 (3.5)	1131 (3.2)	302 (6.1)	0.138	0.001	307 (6.2)	302 (6.1)	0.004
Blood loss anaemia	37 (0.1)	28 (0.1)	9 (0.2)	0.028	0.046	8 (0.2)	9 (0.2)	0.005
Deficiency anaemia	371 (0.9)	259 (0.7)	112 (2.3)	0.126	<0.001	71 (1.4)	112 (2.3)	0.061
Alcohol abuse	241 (0.6)	190 (0.5)	51 (1.0)	0.056	<0.001	77 (1.5)	51 (1.0)	0.046
Drug abuse	69 (0.2)	54 (0.2)	15 (0.3)	0.032	0.026	21 (0.4)	15 (0.3)	0.020
Psychosis	151 (0.4)	112 (0.3)	39 (0.8)	0.064	<0.001	30 (0.6)	39 (0.8)	0.022
Depression	1074 (2.6)	826 (2.3)	248 (5.0)	0.143	<0.001	248 (5.0)	248 (5.0)	0.001
Charlson Comorbidity Index								
0	30,361 (74.8)	28,101 (78.9)	2260 (45.5)	0.744	<0.001	2306 (46.4)	2260 (45.5)	0.035
1–2	7484 (18.4)	5703 (16.0)	1781 (35.8)			1802 (36.3)	1781 (35.8)	
3–4	2186 (5.4)	1441 (4.0)	745 (15.0)			694 (14.0)	745 (15.0)	
5+	571 (1.4)	386 (1.1)	185 (3.7)			169 (3.4)	185 (3.7)	

CHF: congestive heart failure, PVD: peripheral vascular disease, COPD: chronic obstructive pulmonary disease, PSM: propensity score matched, SMD: standardized mean difference. Pearson’s chi-square or Fisher’s exact tests were applied to compare diabetes with variables.

**Table 2 viruses-13-02401-t002:** Comparison and simple logistic regression analysis of ICU admission with diabetes and other variables in unmatched and propensity score matched cohorts.

Variables	Unmatched Cohort				PSM Cohort	
	No ICU Admission (N = 34,634) n (%)	ICU Admission (N = 5968) n (%)	OR (95%CI)	*p*-Value	OR (95%CI)	*p*-Value
Diabetes	4060 (11.7)	911 (15.3)	1.36 (1.25–1.47)	<0.001	1.15 (1.04–1.28)	0.009
Men	17,430 (50.3)	3848 (64.5)	Reference		Reference	
Women	17,204 (49.7)	2120 (35.5)	0.56 (0.53–0.59)	<0.001	0.56 (0.50–0.63)	<0.001
Age groups						
20–29	982 (2.8)	75 (1.3)	Reference		Reference	
30–39	1427 (4.1)	149 (2.5)	1.37 (1.03–1.83)	0.033	0.33 (0.09–1.47)	0.138
40–49	2161 (6.2)	318 (5.3)	1.92 (1.49–2.52)	<0.001	0.60 (0.18–2.39)	0.441
50–59	4456 (12.9)	907 (15.2)	2.66 (2.10–3.43)	<0.001	0.54 (0.17–2.11)	0.348
60–69	5261 (15.2)	1413 (23.7)	3.51 (2.77–4.51)	<0.001	0.65 (0.20–2.54)	0.507
70–79	7838 (22.6)	1826 (30.6)	3.04 (2.41–3.90)	<0.001	0.48 (0.15–1.89)	0.271
80–89	9365 (27.0)	1167 (19.6)	1.63 (1.29–2.09)	<0.001	0.22 (0.07–0.86)	0.032
90+	3144 (9.1)	113 (1.9)	0.47 (0.35–0.64)	<0.001	0.04 (0.01–0.17)	<0.001
Comorbidities						
Myocardial infarction	239 (0.7)	101 (1.7)	2.48 (1.95–3.12)	<0.001	2.27 (1.68–3.04)	<0.001
Cardiac arrhythmias	3612 (10.4)	623 (10.4)	1.00 (0.91–1.09)	0.978	0.93 (0.82–1.05)	0.267
Valvular heart diseases	844 (2.4)	146 (2.5)	1.00 (0.84–1.20)	0.957	0.87 (0.69–1.10)	0.255
Hypertension	8929 (25.8)	1609 (27.0)	1.06 (1.00–1.13)	0.056	1.09 (0.98–1.22)	0.112
Congestive heart failure	1676 (4.8)	335 (5.6)	1.17 (1.04–1.32)	0.012	1.25 (1.08–1.45)	0.004
Peripheral vascular disease	1119 (3.2)	222 (3.7)	1.16 (1.00–1.34)	0.054	1.07 (0.89–1.28)	0.449
Stroke	1024 (3.0)	188 (3.1)	1.07 (0.91–1.25)	0.416	1.01 (0.83–1.22)	0.915
COPD	1854 (5.3)	417 (7.0)	1.33 (1.19–1.48)	<0.001	1.23 (1.05–1.43)	0.010
Pulmonary circulation disorders	436 (1.3)	155 (2.6)	2.09 (1.73–2.51)	<0.001	2.19 (1.65–2.87)	<0.001
Dementia	1975 (5.7)	75 (1.3)	0.21 (0.17–0.26)	<0.001	0.14 (0.09–0.20)	<0.001
Rheumatoid disorders	227 (0.7)	27 (0.5)	0.69 (0.45–1.01)	0.059	0.65 (0.37–1.07)	0.097
Peptic ulcer disease	68 (0.2)	18 (0.3)	1.55 (0.89–2.55)	0.116	1.08 (0.46–2.21)	0.852
Liver disease	848 (2.5)	183 (3.1)	1.26 (1.07–1.48)	0.006	1.15 (0.94–1.40)	0.173
Paralysis	105 (0.3)	21 (0.3)	1.17 (0.71–1.83)	0.524	1.68 (0.92–2.91)	0.091
Other neurological disorders	970 (2.8)	101 (1.7)	0.60 (0.48–0.73)	<0.001	0.56 (0.40–0.75)	<0.001
Renal disease	2741 (7.9)	432 (7.2)	0.91 (0.82–1.01)	0.071	0.81 (0.71–0.93)	0.002
Cancer	1128 (3.3)	190 (3.2)	0.98 (0.83–1.14)	0.774	1.00 (0.78–1.27)	0.999
Hypothyroidism	1157 (3.3)	178 (3.0)	0.89 (0.76–1.04)	0.150	0.93 (0.75–1.14)	0.508
Coagulation disorders	167 (0.5)	85 (1.4)	2.98 (2.29–3.87)	<0.001	4.06 (2.81–5.85)	<0.001
Fluid and electrolyte disorders	1280 (3.7)	153 (2.6)	0.69 (0.58–0.81)	<0.001	0.89 (0.71–1.11)	0.292
Deficiency anemia	328 (0.9)	43 (0.7)	0.76 (0.55–1.04)	0.084	0.72 (0.46–1.09)	0.123
Alcohol abuse	188 (0.5)	53 (0.9)	1.65 (1.20–2.22)	0.002	1.54 (1.01–2.29)	0.046
Psychosis	126 (0.4)	25 (0.4)	1.16 (0.74–1.75)	0.511	1.12 (0.58–1.99)	0.721
Depression	937 (2.7)	137 (2.3)	0.85 (0.70–1.01)	0.065	0.93 (0.72–1.18)	0.552
Charlson Comorbidity Index						
0	25,868 (74.7)	4493 (75.3)	Reference		Reference	
1–2	6383 (18.4)	1101 (18.4)	0.99 (0.92–1.07)	0.851	0.90 (0.80–1.01)	0.083
3–4	1885 (5.4)	301 (5.0)	0.92 (0.81–1.04)	0.188	0.84 (0.72–0.99)	0.035
5+	498 (1.4)	73 (1.2)	0.85 (0.65–1.08)	0.176	0.79 (0.58–1.05)	0.110

CI: confidence interval, COPD: chronic obstructive pulmonary disease, ICU: intensive care unit, OR: odds ratio, PSM: propensity score matched.

**Table 3 viruses-13-02401-t003:** Comparison and simple logistic regression analysis of in-hospital mortality with diabetes and other variables in unmatched and propensity score-matched cohorts.

Variables	Unmatched Cohort				PSM Cohort	
	No In-Hospital Mortality (N = 34,033) n (%)	In-Hospital Mortality (N = 6569) n (%)	OR (95%CI)	*p*-Value	OR (95%CI)	*p*-Value
Diabetes	4034 (11.9)	937 (14.3)	1.24 (1.15–1.34)	<0.001	1.08 (0.97–1.19)	0.146
Men	17,582 (51.7)	3696 (56.3)	Reference		Reference	
Women	16,451 (48.3)	2873 (43.7)	0.83 (0.79–0.88)	<0.001	0.84 (0.76–0.94)	<0.001
Age						
20–29	1048 (3.1)	9 (0.1)	Reference		Reference	
30–39	1557 (4.6)	19 (0.3)	1.41 (0.65–3.31)	0.396	0.14 (0.00–5.63)	0.250
40–49	2444 (7.2)	35 (0.5)	1.65 (0.82–3.68)	0.168	0.12 (0.01–3.42)	0.165
50–59	5173 (15.2)	190 (2.9)	4.20 (2.28–8.92)	<0.001	0.46 (0.09–11.60)	0.533
60–69	6093 (17.9)	581 (8.8)	10.90 (5.99–22.90)	<0.001	1.05 (0.20–25.80)	0.965
70–79	8018 (23.6)	1646 (25.1)	23.50 (12.9–49.20)	<0.001	2.23 (0.43–54.80)	0.402
80–89	7727 (22.7)	2805 (42.7)	41.50 (22.9–87.00)	<0.001	3.75 (0.72–92.40)	0.133
90+	1973 (5.8)	1284 (19.5)	74.40 (40.9–157.00)	<0.001	6.80 (1.30–168.00)	0.019
ICU admission	3991 (11.7)	1977 (30.1)	3.24 (3.04–3.45)	<0.001	3.09 (2.75–3.47)	<0.001
Comorbidities						
Myocardial infarction	219 (0.6)	121 (1.8)	2.90 (2.31–3.62)	<0.001	2.27 (1.69–3.03)	<0.001
Cardiac arrhythmias	3160 (9.3)	1075 (16.4)	1.91 (1.77–2.06)	<0.001	1.78 (1.60–1.99)	<0.001
Valvular heart disease	751 (2.2)	239 (3.6)	1.67 (1.44–1.94)	<0.001	1.62 (1.32–1.96)	<0.001
Hypertension	8737 (25.7)	1801 (27.4)	1.09 (1.03–1.16)	0.003	0.75 (0.68–0.84)	<0.001
Congestive heart failure	1326 (3.9)	685 (10.4)	2.87 (2.61–3.16)	<0.001	2.72 (2.38–3.10)	0.000
Peripheral vascular disease	962 (2.8)	379 (5.8)	2.11 (1.86–2.38)	<0.001	1.78 (1.52–2.09)	<0.001
Stroke	887 (2.6)	325 (4.9)	1.95 (1.71–2.21)	<0.001	1.75 (1.47–2.07)	<0.001
COPD	1815 (5.3)	456 (6.9)	1.32 (1.19–1.47)	<0.001	1.20 (1.03–1.40)	0.022
Pulmonary circulation disorders	457 (1.3)	134 (2.0)	1.53 (1.26–1.85)	<0.001	1.67 (1.24–2.21)	0.001
Dementia	1344 (3.9)	706 (10.7)	2.93 (2.66–3.22)	<0.001	2.32 (2.00–2.68)	<0.001
Rheumatoid disease	205 (0.6)	49 (0.7)	1.24 (0.90–1.68)	0.183	1.37 (0.90–2.03)	0.140
Peptic ulcer disease	66 (0.2)	20 (0.3)	1.58 (0.93–2.56)	0.087	1.33 (0.62–2.61)	0.443
Liver disease	847 (2.5)	184 (2.8)	1.13 (0.96–1.32)	0.144	1.03 (0.84–1.26)	0.782
Paralysis	100 (0.3)	26 (0.4)	1.35 (0.86–2.06)	0.183	0.77 (0.35–1.49)	0.456
Other neurological disorders	774 (2.3)	297 (4.5)	2.04 (1.77–2.33)	<0.001	2.13 (1.72–2.62)	<0.001
Renal disease	2243 (6.6)	930 (14.2)	2.34 (2.15–2.54)	<0.001	2.28 (2.04–2.55)	<0.001
Cancer	965 (2.8)	353 (5.4)	1.95 (1.72–2.20)	<0.001	1.86 (1.51–2.28)	<0.001
Hypothyroidism	1171 (3.4)	164 (2.5)	0.72 (0.61–0.85)	<0.001	0.60 (0.47–0.76)	<0.001
Coagulation disorders	193 (0.6)	59 (0.9)	1.59 (1.18–2.12)	0.003	1.66 (1.09–2.47)	0.020
Fluid and electrolyte disorders	1172 (3.4)	261 (4.0)	1.16 (1.01–1.33)	0.036	1.19 (0.97–1.45)	0.094
Deficiency anaemia	307 (0.9)	64 (1.0)	1.08 (0.82–1.41)	0.567	0.84 (0.55–1.24)	0.394
Alcohol abuse	177 (0.5)	64 (1.0)	1.88 (1.40–2.50)	<0.001	1.44 (0.94–2.14)	0.090
Psychosis	125 (0.4)	26 (0.4)	1.08 (0.69–1.63)	0.714	1.15 (0.61–2.01)	0.651
Depression	907 (2.7)	167 (2.5)	0.95 (0.80–1.12)	0.575	0.84 (0.65–1.07)	0.161
Charlson Comorbidity Index						
0	26,400 (77.6)	3961 (60.3)	Reference		Reference	
1–2	5752 (16.9)	1732 (26.4)	2.01 (1.88–2.14)	<0.001	2.47 (2.18–2.80)	<0.001
3–4	1491 (4.4)	695 (10.6)	3.11 (2.82–3.42)	<0.001	3.99 (3.44–4.62)	<0.001
5+	390 (1.1)	181 (2.76)	3.09 (2.58–3.70)	<0.001	4.40 (3.45–5.59)	<0.001

CI: confidence interval, COPD: chronic obstructive pulmonary disease, OR: odds ratio, PSM: propensity score matched.

## Data Availability

Data included in this study is a property of the Austrian National Public Health Institute (Gesundheit Österreich GmbH). Further information regarding data access is available at: https://datenplattform-covid.goeg.at/english, accessed on 3 May 2021.

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
