# Peer review of "COVID-19 In-Hospital Mortality in People with Diabetes Is Driven by Comorbidities and Age—Propensity Score-Matched Analysis of Austrian National Public Health Institute Data"

_viruses, 2021, doi:10.3390/v13122401_

Round 1
Reviewer 1 Report
This real world study by Dr Faisal Aziz et al. assessed the impact of diabetes on ICU admission and in-hospital death on a countrywide cohort of hospitalized COVID-19 patients using unmatched and PSM logistic regression. The results are generally well presented and discussed; however, the following issues need to be addressed before publication.
- The title is overstated. The reviewer suggests to indicate the use of PSM analysis in the title to avoid possible misunderstanding, and to include a clearer explanation of why the authors chose PSM analysis for assessment in the introduction.
- Page 2, line 54 & Page 8, line 221: Although with a p value over 0.05, the OR of diabetes on in-hospital mortality in PSM cohort remains above 1, which still indicate a tendency of harmful impact. The expression “no significant association” will deliver a more accurate massage than “no association” in this case. The PSM analysis in French patients (doi: 10.1016/j.diabet.2020.101222) which cited in the manuscript also used “no significant association” but not “no association” to describe similar results.
- Page 2, line 83: Lack of references for the sentence “However, evidence concerning the impact... is independent of age, sex, and multimorbidity is still emerging”.
- Page 7, line 191: Incomplete sentence “The association of diabetes with in-hospital mortality (OR: 1.08, 95%CI: 0.97–1.19, p=0.146).”
- Several important publications describing the COVID-19 patients with diabetes (eg. PMIDs: 32409498, 32409499, 34135014) should be compared or at least discussed.
Author Response
Response to Reviewer 1 Comments
This real world study by Dr Faisal Aziz et al. assessed the impact of diabetes on ICU admission and in-hospital death on a countrywide cohort of hospitalized COVID-19 patients using unmatched and PSM logistic regression. The results are generally well presented and discussed; however, the following issues need to be addressed before publication.
Point 1: The title is overstated. The reviewer suggests to indicate the use of PSM analysis in the title to avoid possible misunderstanding, and to include a clearer explanation of why the authors chose PSM analysis for assessment in the introduction.
Response 1: As suggested by the reviewer, we have modified the title as follows; “COVID-19 in-hospital mortality in people with diabetes is driven by comorbidities and age – Propensity score-matched analysis of Austrian National Public Health Institute data”. Also, the justification for choosing the PSM analysis has been included in the introduction (Page 2, Line 88-89).
Point 2: Page 2, line 54 & Page 8, line 221: Although with a p value over 0.05, the OR of diabetes on in-hospital mortality in PSM cohort remains above 1, which still indicate a tendency of harmful impact. The expression “no significant association” will deliver a more accurate massage than “no association” in this case. The PSM analysis in French patients (doi: 10.1016/j.diabet.2020.101222) which cited in the manuscript also used “no significant association” but not “no association” to describe similar results.
Response 2: We agree with the reviewer and therefore have added the text ‘no significant association’ in the suggested text.
Point 3: Page 2, line 83: Lack of references for the sentence “However, evidence concerning the impact... is independent of age, sex, and multimorbidity is still emerging”.
Response 3: As suggested by the reviewer, we have added two references (13,14) in line 83 of the manuscript to support our statement.
Point 4: Page 7, line 191: Incomplete sentence “The association of diabetes with in-hospital mortality (OR: 1.08, 95%CI: 0.97–1.19, p=0.146).
Response 4: We are thankful to the reviewer for pointing out this mistake. We have completed the sentence on Page 7, line 195-196 as follows.
“The association of diabetes with in-hospital mortality (OR: 1.08, 95%CI: 0.97–1.19, p=0.146) became insignificant in the PSM analysis.”
Point 5: Several important publications describing the COVID-19 patients with diabetes (e.g. PMIDs: 32409498, 32409499, 34135014) should be compared or at least discussed.
Response 5: As suggested by the reviewer, we have added the studies by Chen et al., 2020 (PMID: 32409498) on page 9, line 256-257, Gao et al. 2020 (PMID: 32409499) on page 9, line 255, and Ran et al. 2021 (PMID: 34135014) on page 9, line 272-275 in the discussion as well as added the references 27-29.
Reviewer 2 Report
The study assessed the association of COVID-19 severity in the patients who had diabetes and other comorbidities. The study is very interesting and presented very well. COVID-19 diabetic patients suffer much more than patients who do not have diabetes. The study findings are exceptionally significant, and I appreciate the authors for such a magnificent work.
Author Response
Response to Reviewer 2 Comments
The study assessed the association of COVID-19 severity in the patients who had diabetes and other comorbidities. The study is very interesting and presented very well. COVID-19 diabetic patients suffer much more than patients who do not have diabetes.
Point 1: The study findings are exceptionally significant, and I appreciate the authors for such a magnificent work.
Response 1: We are thankful to the reviewer for these positive remarks.
Reviewer 3 Report
This retrospective analysis analysed the impact of diabetes on intensive care unit (ICU) admission and in-hospital mortality in hospitalized COVID-19 patients in Austria. Data were taken from an Austrian data platform and analysed with both unmatched and propensity-score matched (PSM) logistic regression.
The analysis showed a significant higher risk for patients with diabetes (n=4,971) for ICU-admission in both analyses and a higher mortality rate, but only in the unmatched analysis.
The paper is well written and presents important data. This reviewer only has a few minor comments.
1. While matching worked well for most parameters in the PSM cohort, there were some parameters with a fairly high remaining standardised mean difference, in particular hypertension (which is not included in the Charlson Comorbidity Index). The quality of matching and the impact of these parameters should be discussed.
2. Page 7, lines 191/192: incomplete sentence
3. There are some interesting findings in the dataset that might warrant further discussion, even though they are outside the main scope of the paper:
a. Dementia was associated with a significantly lower risk of ICU admission, but had a considerably higher risk of in-hospital mortality in both analyses.
b. Hypertension and hypothyroidism had a significantly lower risk of in-hospital mortality in the PSM cohort.
Author Response
Response to Reviewer 3 Comments
This retrospective analysis analysed the impact of diabetes on intensive care unit (ICU) admission and in-hospital mortality in hospitalized COVID-19 patients in Austria. Data were taken from an Austrian data platform and analysed with both unmatched and propensity-score matched (PSM) logistic regression. The analysis showed a significant higher risk for patients with diabetes (n=4,971) for ICU-admission in both analyses and a higher mortality rate, but only in the unmatched analysis.
The paper is well written and presents important data. This reviewer only has a few minor comments.
Point 1: While matching worked well for most parameters in the PSM cohort, there were some parameters with a fairly high remaining standardised mean difference, in particular hypertension (which is not included in the Charlson Comorbidity Index). The quality of matching and the impact of these parameters should be discussed.
Response 1: We thank the reviewer for raising this point. Indeed, hypertension was not completely balanced even after PSM. To address this issue, we adjusted for the effect of hypertension on the association between diabetes and mortality in the PSM cohort and the results did not change significantly (OR: 1.09, 95%CI: 0.99 – 1.21, p=0.065). We have added a sentence regarding this issue on page 4, line 149-150 and page 7, line 196-198.
Point 2: Page 7, lines 191/192: incomplete sentence
Response 2: We are thankful to the reviewer for pointing out this mistake. We have completed the sentence on Page 7, line 195-196 as follows.
“The association of diabetes with in-hospital mortality (OR: 1.08, 95%CI: 0.97–1.19, p=0.146) became insignificant in PSM analysis.”
Point 3: There are some interesting findings in the dataset that might warrant further discussion, even though they are outside the main scope of the paper:
Point 3a: Dementia was associated with a significantly lower risk of ICU admission, but had a considerably higher risk of in-hospital mortality in both analyses.
Response 3a: We agree with the reviewer’s comment regarding the association of dementia with ICU admission and in-hospital mortality. However, we would like to point out that these are rather crude associations that did not adjust for any other factors and therefore must be interpreted with caution. As acknowledged by the reviewer, our primary objective was to assess the independent impact of diabetes on both outcomes, we therefore did not explore the independent association of each comorbidity with outcomes as it would require either generating a separate PSM cohort for each comorbidity or applying advanced regression methods that can handle a large number of covariates like our study (27 variables). Nevertheless, we have added a statement regarding the dementia on page 9, line 259-262.
Point 3b: Hypertension and hypothyroidism had a significantly lower risk of in-hospital mortality in the PSM cohort.
Response 3b: We agree with the reviewer’s comment regarding the association of hypertension and hypothyroidism with ICU admission and in-hospital mortality. However, we would like to point out that these are rather crude associations that did not adjust for any other factors and therefore these results must be interpreted with caution. As acknowledged by the reviewer, our primary objective was to assess the independent impact of diabetes on both outcomes, we therefore did not explore the independent association of these comorbidity with outcomes as it would require either generating a separate PSM cohort for each comorbidity or applying advanced regression methods that can handle a large number of covariates like our study (27 variables).